# Behavioural Variability in Chicks vs. the Pattern of Behaviour in Adult Hens

**DOI:** 10.3390/ani10020269

**Published:** 2020-02-09

**Authors:** Iwona Rozempolska-Rucińska, Kornel Kasperek, Kamil Drabik, Grzegorz Zięba, Agnieszka Ziemiańska

**Affiliations:** Institute of Biological Basis of Animal Production, University of Life Sciences in Lublin, Akademicka 13, 20-950 Lublin, Poland; iwona.rucinska@up.lublin.pl (I.R.-R.); kornel.kasperek@up.lublin.pl (K.K.); kamil.drabik2@gmail.com (K.D.); grzegorz.zieba@up.lublin.pl (G.Z.)

**Keywords:** behavioural test, breed of laying hens, chick, temperament

## Abstract

**Simple Summary:**

Environmental requirements ensuring behavioural welfare to laying hens may vary depending on the breed. Chickens representing various breeds and reared in the same environment were found not only to differ in the level of activity, emotional arousal, and degree of curiosity, but also to prefer different enrichments of the environment, which was reflected by different levels of stress in these birds. Hence, a question was posed whether the behavioural differences observed were innate behavioural patterns typical of the breed or whether they are an effect of the modifying impact of the environment, which varies between breeds. It has been hypothesised that differences observed already in chicks of different breeds may not be associated with the modifying effect of the environment. Instead, they may be a genetically determined breed-specific behaviour. The present investigations consisted in behavioural tests and assessment of the behaviour of chicks of three laying hen breeds. The study involved 60 green-legged partridge (Zk), 60 Polbar (Pb), and 60 Leghorn (Lg) chicks. The investigations have demonstrated that the birds from the analysed breeds exhibit behavioural differences already on the first days of life. The effect of the breed was evident in the case of such traits as strategy for acquisition of food resources, fearfulness/curiosity, and interest in elements of the environment. With age, chicks may exhibit changes in their emotions, e.g., more pronounced fearfulness, and environmental preferences. However, in the latter case, there is clear tendency towards breed-specific behaviours exhibited from the first days of life. The level of activity, which largely differentiates adult birds, does not discriminate between chicks.

**Abstract:**

The aim of the study was to assess the behaviour of chicks of three different breeds of laying hens differing in the activity, emotional reactivity, and environmental preferences. Another objective was to answer the question whether the behavioural differences between adult birds would be evident already in the chick period or whether they are an effect of the further modifying impact of the environment. 60 green-legged partridge, 60 Polbar, and 60 Leghorn chicks were used in the experiments. The chicks hatched in a flock where hens were previously assessed with behavioural tests and the corticosterone levels in their feathers was determined, indicating significant differences in the temperament and stress level between the breeds. Five tests were carried out: two on competitiveness, activity, interest and fearfulness/curiosity. The experiments revealed considerable differences between the chicks. The Zk birds coped better with situations requiring swiftness and initiative. The Pb chicks were slower than Zk and Lg and did not make quick decisions. Hence, a lower number of these birds entering and leaving the test cage and staying inside was recorded. The Zk chicks exhibited a higher level of fearfulness than the other breeds. In terms of the environment enrichment elements, sand and woodchips were more attractive to the Zk chicks, whereas the Lg and Pb birds preferred pecking the string. No differences in the time of undertaking the analysed activities were found between the breeds.

## 1. Introduction

To ensure birds’ welfare, their housing environment has to be adjusted to not only biological but also behavioural needs. The following questions associated with the welfare and housing conditions provided to chickens arise: (i) Is the genotype of birds suitable for each type of environment? and (ii) is it necessary to adjust the environment (rearing practices) to the genotype? [1,2]. It appears that birds’ environmental needs may vary not only between species but also between breeds [3] and individuals of the same breed [4]. Previous investigations [3,5] have demonstrated that hens from three different breeds (green-legged partridge—Zk, Polbar—Pb, and Leghorn—Lg) kept in environments as identical as possible not only differed in the level of activity, emotional arousal, or degree of curiosity, but also preferred different enrichments of the environment, which collectively resulted in a different level of stress in the birds [5]. The Zk breed showed interest in objects that allowed scratching and searching, i.e., the birds preferred a container filled with finely shredded straw and woodchips as well as insect larvae but ignored a feed container, shelter box, or mirror. Simultaneously, the breed was assigned the lowest level of stress. The Lg chicks were mainly interested in the mirror and observation of their reflection, whereas the Pb chickens did not exhibit specific preferences, although they tended to be attracted to the sand contained. Both breeds had a higher level of stress than the Zk chicks. The authors reported that the living environment fulfilled the needs of the Zk breed only, but failed to satisfy the needs of the Lg hens [3,5]. The birds were kept on bedding and thus the need of the Zk chickens to scratch and search was satisfied. 

Since the behavioural patterns depend on the genetic background, previous bird’s experiences, environmental conditions prevailing during embryonic development, and epigenetic effects [4,6,7,8,9], a question was posed whether the differences observed in the behaviour of the three breeds (Zk, Lg, Pb) were innate breed-specific behavioural patterns or whether they were “generated” by their living environment. In other words, the question was whether the rearing practice, which was not adjusted to the genetic predisposition of the breed, was the cause of the behavioural differences or whether the breed-specific differences occurred regardless of the rearing practices. An environment that prevents expression of behaviours to which the animal is motivated may evoke other variants of behaviour, often leading to behavioural problems [10]. These issues are highly important for an adequate choice of birds suitable for various rearing systems, where genetic-environmental interactions may result in problems with welfare and, consequently, productivity [1,2,11].

The research hypothesis assumes that any differences in the behaviour between young chicks of different breeds cannot result from the modifying impact of the environment at such an early age.

Behavioural variability in this case is associated with innate behavioural patterns. In turn, if the differences in the behaviour between chicks of different breeds are insignificant, it can be assumed that the behaviour of adult hens is an effect of environmental factors.

The aim of the study was to assess the behaviour of chicks representing three different laying hen breeds whose adults differ in the activity, emotional reactivity, and environmental preferences. Another objective was to answer the following questions:Will the differences be evident already in the chick period as breed-specific traits?Will the chicks exhibit no behavioural differences? This would indicate that differences in behaviour visible in adult birds are not breed-specific patterns but are associated by the modifying effect of the environment.

## 2. Materials and Methods 

All procedures employed during the research were approved by the II Local Ethics Committee for Animal Testing at the University of Life Sciences in Lublin, Poland (Approval No. 69/2017 of 28 September 2017). Chicks of three breeds of laying hens were used in the experiments: 60 green-legged partridge (Zk), 60 Polbar (Pb), and 60 Leghorn (Lg) birds. The chicks hatched in a flock where hens were previously assessed with behavioural tests and the corticosterone levels in their feathers was determined, indicating significant differences in the temperament and stress level between the breeds [3,5]. 

The native Zk breed with colourful plumage is often reared in organic farming systems. These hens are perfectly adapted to the conditions of extensive free-range farming. Pb is an autosexing hybrid that has emerged via mating of Green-legged partridge hens with Plymouth Rock cocks. The birds of this breed have dappled plumage. Lg birds with white plumage are one of the most popular breeds of laying hens in Europe. It is extremely well adapted to intensive rearing. Lg hens lay approximately 230 eggs each with a weight of 65.5 g. The laying performance of Zk and Pb hens, which are kept in conservative non-selected flocks, is substantially lower, i.e., 160 eggs with an average weight of 45 g. The body weight of 18-week-old birds is Pb-1350g, Zk-120g, and Lg-1300g (data provided by inspection of the laying performance of the analysed flocks). 

One-day-old chicks were placed in 6 pens. There were 10 individuals of each breed in each pen, i.e., 30 chicks in total. The group of birds within the pen did not change from the 1st to 23rd day of life. The only exception was one of the pens where a Leghorn chick was replaced on the 2nd day of life due to health condition. The pens with a surface area of 100 cm × 70 cm (210 cm^2^ of free space per chick) and 60-cm high walls were kept in an experimental room at controlled temperature (22 ℃) and humidity (68%). Each pen was heated with a 250W infrared heater controlling the temperature of the pen floor, i.e., 31 ℃ in the case of the one-day-old chicks. The temperature was gradually reduced to 23 ℃ on the 21st day of life. Each pen was equipped with two round drinkers with a 20-cm diameter and two 30-cm long feeders adapted for chicks. Complete diet and water were available ad libitum. The chicks were randomly selected for the experiment from a group of 500 chicks that had hatched on the Experimental Farm of the Scientific Institute. The birds were randomly allocated in the pens. The chicks from the different breeds differ in the colour of fluff, which facilitated identification thereof during subsequent experiments. The body weight of the chicks ranged from 35 to 37 g and did not differ significantly between the breeds. The five tests employed in the experiment are described in detail in Table 1. Each test was recorded with a digital camera. In total, 20-h recordings of 10-min tests and 36-h recordings of 30-min tests were analysed. The tests were carried out once a day from 9:00 to 16:00, each time starting with a different pen. The tests were started at different times of the day and in different order to exclude behavioural conditioning. The next test on the same day was conducted only when the chicks from all pens exhibited chick ethogram-specific behaviours, i.e., rest, feed and water ingestion, and exploration.

The recorded indicators did not have a normal distribution; hence, the data was subjected to rank transformation. Multiple comparisons with the Bonferroni correction of the estimates of differences in the examined traits between the breeds were analysed in two-factor models, taking into account the effect of the genetic group and the age of chicks at the time of the experiment. The number of the pen where the chicks were kept was not a significant factor and was therefore not included in the analyses. The GLIMMIX procedure (SAS Institute, Cary, NC, USA) was applied. The results are presented for the chicks on the first and last days of the experiment and for the entire test period. 

## 3. Results

The competitiveness of the chicks was assessed by recording the number of birds from the analysed breeds staying inside the cage with feed, entering and leaving the cage at a specific time, and staying next to the cage with a clear indication of the willingness to enter the cage (Table 2).

The largest group of three-day-old chicks present inside the cage was represented by the Pb breed. However, the lowest number of Pb chicks was noted inside the cage in the three -week-old group.

Noteworthy, there were no significant differences in the number of the three -day-old chicks of the respective breeds entering, leaving, and staying next to the cage. Such differences were recorded at the age of three weeks: the greatest number of chicks entering and leaving the cage represented the Zk and Lg breeds, and Pb chicks accounted for the lowest number. 

The assessment of the numbers of birds entering and leaving the cage showed no significant differences between Zk and Lg on day 3 and day 21. In turn, in terms of the total measurements in the experimental period, the Zk chicks represented the biggest number in the observations of each of the traits.

In the next competitiveness test (II), the time to approach to the nettle leaf and the number of pecks were evaluated (Table 3). Zk appeared to be the most responsive breed in comparison with the other chicks. There were no significant differences between the breeds only in one case, i.e., the time of holding of the nettle in the beak did not differ between the four-day-old Zk and Pb chicks. The Zk chicks exhibited the greatest interest in the object measured by the number of pecks. The time of holding of nettle in the beak and the number of pecks did not differ between the Pb and Lg breeds, regardless of the age.

Summing up the competitiveness test, the Zk breed scored the highest results in most of the tests. The results changed with the age of the chicks and between the indicators analysed.

Another analysed trait was activity (Table 4).

The chicks did not differ significantly in the level of activity throughout the experimental period.

Additionally, differences in chicks’ preferences for environmental elements were estimated in the tests (Table 5).

The sand and woodchips turned out to be especially attractive to the Zk chicks. The 4-day-old Zk chicks chose the container with sand more often than Pb and Lg, whereas differences in the interest in the woodchips were observed in three-week-old chicks. There were no differences between the number of Lg and Pb showing interest in these objects, irrespective of their age. In contrast to Zk, the Lg and Pb birds were interested in the string. The Lg chicks showed interest in this object mainly in early life. There were no differences between the breeds in the older chicks. 

Another trait examined in the experiment was fearfulness/curiosity (Table 6). Significant differences between the breeds were noted in the tests. The Zk chicks were characterised by a substantially higher level of fearfulness than Pb and Lg. This was already evident on the first days of life and, despite the continuous positive contact with humans (feeding, treats), this trend did not change until the end of the experiment. On the first days of life, there were no differences in fearfulness between Pb and Lg. However, the older Lg chicks tended to avoid stimuli and the number of birds approaching the hand was significantly lower than in the case of Pb.

The analysis of the “direct contact with the hand” parameter demonstrated that the level of fearfulness changed with age. In the group of the three-day-old chicks, there were no significant differences, or they were on the border of significance (Lg-Zk), whereas highly significant differences in the level of fearfulness/curiosity were noted in the three-week-old chicks. Similarly, in the “approach to the hand” test, the highest level of avoidance of contact was noted in the Zk breed. The Pb breed exhibited the lowest level of fearfulness. During the test, the chicks would fall asleep in the palm of the hand and these were not sporadic cases.

There were no differences in the time of undertaking individual activities between the breeds (Table 7).

## 4. Discussion

As demonstrated in many studies, the level of additive variability is so high that selection for modification of chicken’s temperament may bring positive effects [6,7,12]. Currently, behavioural selection is mainly targeted at reduction of aggression and pterophagy [13,14]. Therefore, investigations mainly focus on adjustment of birds’ temperament to the environment. However, as indicated by the present study and previous investigations conducted by the authors [3,5], there is another aspect, i.e., the environment is not universal for individual genotypes and determines hens’ behaviour in various ways.

The analysis of competitiveness test I showed the highest number of Zk chicks staying inside as well as entering and leaving the cage, in comparison with the other two breeds. This test shows not only the level of competitiveness but also the strategy of behaviour, i.e., the chicks entered and stayed in the cage or left the cage to return instantly. The Zk chicks exhibited better performance in situations requiring swiftness and initiative. The Pb birds were slower than the Zk and Lg chicks and did not make decisions as readily, hence their lower number staying inside the cage as well as entering and leaving the cage. These differences were mainly evident in the older chicks (day 21). The Zk chicks turned out to be fast and active and definitely outcompeted the other breeds in both competitiveness tests (I and II). Importantly, Zk chicks come from parents with a lower corticosterone level in the feathers, in comparison with the Lg and Pb breeds [5]. Chicks’ behaviour is modified already at the stage of embryonic development through transfer of hormones between the hen and the egg and the formation of the HPA axis involved in responses to stimuli [15,16,17]. As demonstrated in various studies, chicks with lower competitiveness and increased fearfulness hatch from eggs laid by hens with higher plasma corticosterone levels [8,18,19]. 

The competitiveness tests (I and II) demonstrated significant differences between the breeds depending on the age of the chicks. However, this was not a linear relationship, as competitiveness did not increase or decrease with age. Due to the limited volume of the paper, the tables show the results of tests conducted in several-day-old and several-week-old chicks (at the beginning and end of the experiment). Throughout the entire period, the differences between the competitiveness results varied and one breed had higher results at one time point but lower at another. For instance, there were no differences between the numbers of Zk and Lg chicks staying inside, entering, and leaving the cage on days 3 and 21, but such differences were found in the entire experimental period. This is highly important for inference of the competitiveness of the breeds. Although the Zk breed seemed to exhibit the highest level of competitiveness, as it outcompeted the other breeds in both competitiveness tests (I and II), we believe that this temperament trait is mainly associated with motivation. Importantly, the present competitiveness tests were based on access to feed (treats available in the cage, nettle leaves), and the intention to reach the goal is associated with motivation. Therefore, such tests mainly assess animal’s motivations to take specific actions. Motivation and related emotions are the driving force in undertaking activity [20,21]. Noteworthy, Zk is a primitive breed specialised in search for food. These birds are eagerly kept on organic farms, as they are largely self-sufficient in satisfy their nutritional needs [22], and food search is undoubtedly their strong motivation. 

A characteristic trait in this breed is the behaviour strategy, i.e., the prompt reaction and decision making, which combined with appropriate motivation allowed the Zk chicks to outcompete the other breeds. A breed-specific behaviour strategy can be assumed, but the level of breed competitiveness cannot be clearly determined, as this trait is significantly modified by animal’s motivation. In our opinion, the results of the competitiveness tests were determined by the combination of a specific chick behaviour strategy and motivation.

The analysis of the activity (Table 4) of the chicks revealed quite surprising results, in comparison with our previous investigations of adult birds [3]. The study conducted by Kozak et al. [3] showed significantly higher activity of adult Lg birds than Zk and Pb. Within the same time, Lg chickens examined twice as many objects and crossed a significantly larger number of squares in the open field test [3,5]. The tests applied to the chicks did not demonstrate differences in the activity between the breeds. Therefore, the increased activity of adult Lg chickens may be a result of environmental-induced behavioural modifications. Investigations conducted by Branciari et al [23] clearly showed a change in birds’ behaviour (Lg) depending on the rearing system. As demonstrated by Kozak et al. [3,5], the behavioural needs of Lg chickens may not be fulfilled in the traditional rearing system, which may result in the “evolution” of a different behavioural system that will somehow compensate for animals’ needs [24]. Access to an enriched environment at a young age exerts a positive effect on the development of brain and behaviour in adult animals [25]. In contrast, the impossibility to fulfil species/breed-specific behavioural needs leads to inappropriate development of dendrites in the brain, which results in excessive activity and excitability [21,25]. The theory of improper development of dendrites in the Lg breed should certainly be thoroughly verified; nevertheless, it has been established that these birds cannot satisfy one of their needs, i.e., curiosity, in farm conditions [3,5].

Many studies associate the excessive reactivity and fearfulness of the Lg breed with its white plumage [26,27]. The investigations conducted by Fraisse and Cockrem [28] confirmed this correlation also at the hormonal level, i.e., they showed higher concentrations of corticosterone in birds with white than brown plumage. However, the present analyses do not suggest that the excessive activity of adult Lg birds is characteristic for the breed. The chicks of the three tested breeds did not differ in the type or time of undertaking the activities observed (Table 7). This confirms that the excessive activity of adult Lg hens may be a result of modifying environmental effects.

An important task in the present study was to determine whether the chicks, likewise adult birds, would exhibit differences in environmental preferences. The sand and woodchips were especially attractive to the Zk chicks. This confirms the characteristic element in the behaviour of the breed, i.e., the need for scratching and searching. The Zk breed is popular on family farms and on organic farms, where birds have large areas at their disposal and cope well with searching for food. This behaviour is motivated by a strong instinct. The four-day-old Zk chicks chose the sand container more often than the Pb and Lg birds, while three-week-old chicks exhibited greater interest in the woodchips. With its structure, the latter material is better for scratching and searching than sand; therefore, the older Zk chicks probably preferred the woodchips. There were no differences between the number of Lg and Pb showing interest in these objects, regardless of their age. These breeds, unlike Zk, showed interest in the string. More Lg birds were interested in this object, but only in the early stages of life. There were no differences between the breeds in the older chicks. This may be related to the fact that the chicks “became bored” with the object.

The present study also confirms that fearfulness is one of the temperament traits differentiating the breeds from the first days of life (Table 6). This trait in animals is characterised by a high inheritance rate, which has also been confirmed in laying hens [6,7]. The study has shown that fearfulness changes over time, exhibiting a downward trend in older chicks, which may be associated with habituation to a given stimulus. The neuroendocrine system develops since the moment of hatching; with age, the activity of the HPA axis decreases, which is accompanied by a decline in the expression of the behavioural response to stress [29].

The trends observed in the individual breeds of chicks are visible in adult birds (authors’ observations during the daily care of the birds). The Zk chicks were characterized by the highest level of fearfulness. As indicated by the daily observations on the farm, adult Zk birds have a tendency to avoid humans, and closer contact sometimes results in aggression, mainly in male individuals (authors’ observations). This phenomenon may be associated with the origin of the breed. Zk is a primitive breed with many characteristics of its wild ancestors [22,30]. In comparison with commercial breeds, a distinctive trait of primitive breeds is the clear and more complex stress response [31] with a faster return to homeostasis [32]. This type of behavioural response allows e.g., effective avoidance of predators in the free-range by farmed primitive breeds and those living in a similar environment as their wild ancestors.

Analysis of the level of fearfulness in chicks should consider the fact that the Lg chicks exhibited an intermediate level of this trait between Zk and Pb. The three-day-old Lg chicks were characterised by similar fearfulness to Pb. However, already on day 21, they showed a significantly higher fearfulness degree, which can also be observed in adult farmed birds (own observations). Adult Lg chickens are extremely shy and reactive birds.

## 5. Conclusions

To sum up the present investigations, it can be concluded that birds of different breeds exhibit behavioural differences from the first days of life. The effect of the breed was evident in such features as the strategy for acquisition of food resources, fearfulness/curiosity, and interest in elements of the environment. In the case of environmental preferences, there is a clear tendency to show behaviours characteristic for adult birds from the first days of life. An example of such behaviour in the case of the Zk breed is the need to scratch and search. The level of activity significantly differentiating adult birds does not differentiate chicks. It is important to verify environmental factors that induce the high degree of excitability and activity in adult Lg chickens in subsequent studies. At this stage of research, it can be suggested that inadequate adjustment of the environment to the Lg breed, which exhibits, e.g., high curiosity, can change birds’ behaviour, resulting in the hyperactivity of adults.

## Figures and Tables

**Table 1 animals-10-00269-t001:** Description of the experiment and estimators of chicks’ behaviour.

Test	Course of the Experiment	Indicator/Measurement Unit	Day of Measurement (Age in Days)	Measurement	Duration of the Test (Min)	Definition
Competitiveness I	A cage with openwork walls and an outlet allowing simultaneous entry/exit of only 1 chick was placed into the pen where the chicks were kept. The cage was placed in the pen at different times of the day. It contained feed enriched with treats.	Number of birds inside the cage/birds	3, 6, 9, 12, 15, 18, 21	every 1 min	10	Number of chicks of each breed inside the cage and the time of entry of the first chick of each breed
		Number of birds entering the cage/ birds		Observation throughout the test		Number of chicks of each breed entering the cage
		Number of birds leaving the cage/birds		Observation throughout the test		Number of chicks of each breed leaving the cage
		Number of birds present next to the cage/ birds		every 1 min		Number of birds of each breed staying next to the cage walls
Competitiveness II	The chicks were given a fragment of a nettle leaf with the midrib, which prevented them from eating thereof	Time of holding the nettle leaf in the beak/seconds	4, 7, 10, 13, 16, 20	Observation throughout the test	10	The duration of holding the nettle leaf in the beak by the chicks of each breed was recorded
		Number of pecks of the nettle leaf/number		Observation throughout the test		The number of pecks of the leaf by each chick was recorded
Activity/birds	The chicks were allowed to undertake any activity with no interference of any of the experimental factors	-	4, 7, 10, 13, 16, 20	every 1 min	30	The number of active chicks of each breed was recorded. “Active” meant a chick exhibiting any motor activity
Interest	A container with sand and a container with woodchips were placed in the pen where the chicks were kept. Both containers were placed apart from each other to spot which one was chosen by the chicks	Interest in the sand/ birds	4, 7, 10, 13, 16, 20	every 1 min	30	The number of birds of each breed in the containers with sand and woodchips was recorded
		Interest in woodchips/birds				
	A cardboard square with a red spot was placed on the wall of the pen	Interest in the cardboard square/birds				The number of birds of each breed that approached and pecked the cardboard square was recorded
	A cotton string was hung in the pen	Interest in the cotton string/birds				The number of birds of each breed that approached and pecked the string was recorded
Fearfulness/curiosity	A board (20 cm × 20 cm) with a marked centre was placed in the pen, on which the observer placed his/her hand. The hand did not move	Approach to the hand/birds	3, 6, 9, 12, 15, 18, 21	every 1 min	10	The number of birds of each breed that stepped on the board was recorded
		Direct contact with the hand/birds				The number of birds of each breed that stepped on the hand or pecked the hand was recorded

**Table 2 animals-10-00269-t002:** Mean level of analysed indicators * and significance of differences between the means in the Competitiveness I test in relation to the breed and age of chicks.

Test	Trait	Breed	Estimate *	Age (Days)	Differences	Pr > |t|
Competitiveness I	Number of birds inside the cage	Lg	574.17	3	Lg vs Pb	0.000
		Pb	856.19	3	Pb vs Zk	0.000
		Zk	560.82	3	Lg vs Zk	0.817
		Lg	480.44	21	Lg vs Pb	0.048
		Pb	366.37	21	Pb vs Zk	0.001
		Zk	566.60	21	Lg vs Zk	0.135
		Lg	582.31	total (3–21)	Lg vs Pb	0.281
		Pb	605.80		Pb vs Zk	0.000
		Zk	703.39		Lg vs Zk	0.000
	Number of birds entering the cage	Lg	555.13	3	Lg vs Pb	0.719
		Pb	577.13	3	Pb vs Zk	0.800
		Zk	592.63	3	Lg vs Zk	0.539
		Lg	659.00	21	Lg vs Pb	0.009
		Pb	498.40	21	Pb vs Zk	0.000
		Zk	744.93	21	Lg vs Zk	0.160
		Lg	639.21	total (3–21)	Lg vs Pb	0.000
		Pb	552.90		Pb vs Zk	0.000
		Zk	699.39		Lg vs Zk	0.009
	Number of birds leaving the cage	Lg	550.71	3	Lg vs Pb	0.766
		Pb	532.88	3	Pb vs Zk	0.453
		Zk	577.88	3	Lg vs Zk	0.651
		Lg	684.04	21	Lg vs Pb	0.003
		Pb	506.27	21	Pb vs Zk	0.000
		Zk	745.61	21	Lg vs Zk	0.305
		Lg	649.13	total (3–21)	Lg vs Pb	0.000
		Pb	536.70		Pb vs Zk	0.000
		Zk	705.70		Lg vs Zk	0.013
	Number of birds present next to the cage	Lg	669.38	3	Lg vs Pb	0.870
		Pb	659.08	3	Pb vs Zk	0.349
		Zk	600.26	3	Lg vs Zk	0.271
		Lg	412.81	21	Lg vs Pb	0.845
		Pb	400.51	21	Pb vs Zk	0.001
		Zk	616.88	21	Lg vs Zk	0.001
		Lg	583.60	total (3–21)	Lg vs Pb	0.026
		Pb	636.46		Pb vs Zk	0.141
		Zk	671.45		Lg vs Zk	0.000

* rank-transformed means; Lg, Leghorn; Pb, Polbar; Zk, Green-legged Partridge; Pr > |t|: significance level.

**Table 3 animals-10-00269-t003:** Mean level of analysed indicators* and significance of differences between the means in the Competitiveness II test in relation to the breed and age of chicks.

Test	Indicator	Breed	Estimate*	Age (days)	Differences	Pr > |t|
Competitiveness II	Time of holding the nettle in the beak (seconds)	Lg	189.89	4	Lg vs Pb	0.577
		Pb	209.55	4	Lg vs Zk	0.041
		Zk	256.53	4	Pb vs Zk	0.066
		Lg	244.54	20	Lg vs Pb	0.498
		Pb	273.80	20	Lg vs Zk	0.000
		Zk	380.04	20	Pb vs Zk	0.001
		Lg	253.11	total (4–20)	Lg vs Pb	0.556
		Pb	234.03		Lg vs Zk	0.001
		Zk	322.43		Pb vs Zk	0.038
	Number of pecks	Lg	341.67	4	Lg vs Pb	0.576
		Pb	358.05	4	Lg vs Zk	0.000
		Zk	456.06	4	Pb vs Zk	0.001
		Lg	182.00	20	Lg vs Pb	0.186
		Pb	222.22	20	Lg vs Zk	0.000
		Zk	363.85	20	Pb vs Zk	0.000
		Lg	257.90	total (4–20)	Lg vs Pb	0.683
		Pb	265.67		Lg vs Zk	0.000
		Zk	386.93		Pb vs Zk	0.000

* rank-transformed means; Lg: Leghorn; Pb: Polbar; Zk: Green-legged Partridge; Pr > |t|: significance level.

**Table 4 animals-10-00269-t004:** Mean level of analysed indicators* and significance of differences between the means in the Activity test in relation to the breed and age of chicks.

Test	Breed		Estimate*	Age (days)	Differences	Pr > |t|
Activity (no.)	Lg		463.24	4	Lg vs Pb	0.999
	Pb		463.32	4	Lg vs Zk	0.113
	Zk		533.07	4	Pb vs Zk	0.113
	Lg		415.86	20	Lg vs Pb	0.823
	Pb		425.71	20	Lg vs Zk	0.332
	Zk		458.56	20	Pb vs Zk	0.455
	Lg		532.81	total (4–20)	Lg vs Pb	0.550
	Pb		519.64		Lg vs Zk	0.100
	Zk		569.04		Pb vs Zk	0.025

* rank-transformed means; Lg: Leghorn; Pb: Polbar; Zk: Green-legged Partridge; Pr > |t|: significance level; no: number.

**Table 5 animals-10-00269-t005:** Mean level of analysed indicators* and significance of differences between the means in the Interest test in relation to the breed and age of chicks.

Test	Object (Measurement)	Breed	Estimate *	Age (Days)	Differences	Pr > |t|
Interest	Sand (no.)	Lg	453.19	4	Lg vs Pb	0.361
		Pb	413.13	4	Lg vs Zk	0.031
		Zk	548.07	4	Pb vs Zk	0.002
		Lg	557.31	20	Lg vs Pb	0.064
		Pb	476.05	20	Lg vs Zk	0.074
		Zk	478.89	20	Pb vs Zk	0.948
		Lg	539.02	total (4–20)	Lg vs Pb	0.433
		Pb	521.79		Lg vs Zk	0.324
		Zk	560.69		Pb vs Zk	0.077
	Woodchips (no.)	Lg	353.34	4	Lg vs Pb	0.240
		Pb	401.27	4	Lg vs Zk	0.167
		Zk	409.72	4	Pb vs Zk	0.836
		Lg	506.13	20	Lg vs Pb	0.559
		Pb	482.34	20	Lg vs Zk	0.001
		Zk	641.58	20	Pb vs Zk	0.000
		Lg	521.23	total (4–20)	Lg vs Pb	0.925
		Pb	523.14		Lg vs Zk	0.006
		Zk	577.12		Pb vs Zk	0.008
	Cardboard square (no.)	Lg	29.50	4	Lg vs Pb	0.249
		Pb	46.66	4	Lg vs Zk	0.779
		Zk	34.17	4	Pb vs Zk	0.168
		Lg	29.50	20	Lg vs Pb	0.811
		Pb	34.54	20	Lg vs Zk	0.476
		Zk	44.83	20	Pb vs Zk	0.239
		Lg	46.67	total (4–20)	Lg vs Pb	0.394
		Pb	37.11		Lg vs Zk	0.591
		Zk	40.22		Pb vs Zk	0.612
	String (no.)	Lg	395.49	4	Lg vs Pb	0.023
		Pb	322.10	4	Lg vs Zk	0.000
		Zk	264.49	4	Pb vs Zk	0.032
		Lg	315.05	20	Lg vs Pb	0.942
		Pb	317.42	20	Lg vs Zk	0.741
		Zk	326.36	20	Pb vs Zk	0.709
		Lg	360.10	total (4–20)	Lg vs Pb	0.146
		Pb	333.56		Lg vs Zk	0.015
		Zk	313.21		Pb vs Zk	0.161

*rank-transformed means; Lg: Leghorn; Pb: Polbar; Zk: Green-legged Partridge; Pr > |t|- significance level; no.: number.

**Table 6 animals-10-00269-t006:** Mean level of analysed indicators* and significance of differences between the means in the Fearfulness/curiosity test in relation to the breed and age of chicks.

Test	Indicator	Breed	Estimate*	Age (Days)	Differences	Pr > |t|
Fearfulness/curiosity	Approach to the hand (no.)	Lg	382.72	3	Lg vs Pb	0.383
		Pb	345.53	3	Lg vs Zk	0.002
		Zk	250.89	3	Pb vs Zk	0.027
		Lg	571.52	21	Lg vs Pb	0.005
		Pb	692.58	21	Lg vs Zk	0.000
		Zk	390.87	21	Pb vs Zk	0.000
		Lg	502.67	total (3–21)	Lg vs Pb	0.000
		Pb	563.65		Lg vs Zk	0.000
		Zk	343.68		Pb vs Zk	0.000
	Direct contact with the hand (no.)	Lg	413.35	3	Lg vs Pb	0.280
		Pb	369.73	3	Lg vs Zk	0.045
		Zk	332.42	3	Pb vs Zk	0.356
		Lg	556.85	21	Lg vs Pb	0.000
		Pb	717.54	21	Lg vs Zk	0.000
		Zk	407.23	21	Pb vs Zk	0.000
		Lg	478.52	total (3–21)	Lg vs Pb	0.000
		Pb	564.11		Lg vs Zk	0.000
		Zk	367.37		Pb vs Zk	0.000

* rank-transformed means; Lg: Leghorn; Pb: Polbar; Zk: Green-legged Partridge; Pr > |t|: significance level; no.: number.

**Table 7 animals-10-00269-t007:** Mean time of undertaking activity * and significance of differences between the means in relation to the breed and age of chicks.

Measurement	Breed	Estimate *	Age (Days)	Differences	Pr > |t|
Time before entering the cage (seconds)	Lg	98.33	3	Lg vs Pb	0.239
	Pb	79.42	3	Lg vs Zk	0.704
	Zk	92.25	3	Pb vs Zk	0.423
	Lg	38.58	21	Lg vs Pb	0.316
	Pb	54.67	21	Lg vs Zk	0.992
	Zk	38.42	21	Pb vs Zk	0.311
	Lg	59.27	total (3–21)	Lg vs Pb	0.081
	Pb	69.89		Lg vs Zk	0.733
	Zk	61.33		Pb vs Zk	0.159
Time before entering the sand container (seconds)	Lg	47.92	4	Lg vs Pb	0.955
	Pb	47.25	4	Lg vs Zk	0.295
	Zk	35.50	4	Pb vs Zk	0.321
	Lg	23.08	20	Lg vs Pb	0.949
	Pb	23.83	20	Lg vs Zk	0.877
	Zk	21.25	20	Pb vs Zk	0.827
	Lg	36.88	total (3–21)	Lg vs Pb	0.700
	Pb	39.15		Lg vs Zk	0.565
	Zk	33.48		Pb vs Zk	0.339
Time before entering the woodchip container (seconds)	Lg	58.17	4	Lg vs Pb	0.486
	Pb	50.92	4	Lg vs Zk	0.538
	Zk	51.75	4	Pb vs Zk	0.936
	Lg	20.42	20	Lg vs Pb	0.433
	Pb	28.58	20	Lg vs Zk	0.974
	Zk	20.08	20	Pb vs Zk	0.415
	Lg	35.71	total (3–21)	Lg vs Pb	0.683
	Pb	37.83		Lg vs Zk	0.962
	Zk	35.96		Pb vs Zk	0.718
Time before approach to the hand (seconds)	Lg	82.17	3	Lg vs Pb	0.714
	Pb	76.25	3	Lg vs Zk	0.861
	Zk	85.00	3	Pb vs Zk	0.588
	Lg	27.83	21	Lg vs Pb	0.159
	Pb	50.67	21	Lg vs Zk	0.653
	Zk	35.08	21	Pb vs Zk	0.335
	Lg	53.28	total (3–21)	Lg vs Pb	0.868
	Pb	54.38		Lg vs Zk	0.697
	Zk	55.85		Pb vs Zk	0.823

* rank-transformed means; Lg: Leghorn; Pb: Polbar; Zk: Green-legged Partridge; Pr > |t|: significance level.

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
