# Peer review of "Behavioural Variability in Chicks vs. the Pattern of Behaviour in Adult Hens"

_animals, 2020, doi:10.3390/ani10020269_

Round 1

Reviewer 1 Report

The authors examined behavior of 3 breeds of layer hens in various behavioral tests.

In general, the language needs heavy editing; the phrasing sometimes makes it difficult to understand what the authors mean.

Line 47: change to " kept in environments as identical as possible"

Lines 59-60: "The animal's living environment...." this seems to be an unnecessary sentence

Lines 66-67: "The research hypothesis assumed that the potential differences in the behaviour of chicks representing different breeds cannot result from the modifying impact of the environment." I suppose that is a possible hypothesis, but it seems very unlikely that behaviors from different breeds cannot be the result of modifying impact of environment- as the authors also say in the paragraph before.

Line 72-75: I don't understand the final sentence of the introduction.

Line 97-98: "In total, 20-h recordings of 10-min tests and 36-h recordings of 30-min tests were analysed. " I don't understand this sentence.

Results: I am missing the absolute values of what was measured. It is unclear to me how the comparison values were calculated, and I would like to be able to see whether this is based on a few chicks performing the behavior or many. This makes it very difficult for the reader to interpret results. I am also skeptical about the statistical analysis, which from the tables appears to be one-on-one comparisons of the groups, without reporting overall effects of breed or age.

Discussion: as the results are difficult to interpret, the discussion is as well.

In general it appears that the authors have performed a considerable amount of work; I hope that in revision the paper will be more clear so that these results can be better interpreted.

Author Response

Thank you for the review of the manuscript. Below we address each comment. Changes introduced in response to the review are marked in red in the manuscript.

Początek formularza

Open Review

English language and style

(x) Extensive editing of English language and style required 
( ) Moderate English changes required 
( ) English language and style are fine/minor spell check required 
( ) I don't feel qualified to judge about the English language and style 

Yes

Can be improved

Must be improved

Not applicable

Does the introduction provide sufficient background and include all relevant references?

( )

( )

(x)

( )

Is the research design appropriate?

( )

( )

(x)

( )

Are the methods adequately described?

( )

( )

(x)

( )

Are the results clearly presented?

( )

( )

(x)

( )

Are the conclusions supported by the results?

( )

( )

(x)

( )

Comments and Suggestions for Authors

The authors examined behaviour of 3 breeds of layer hens in various behavioural tests.

In general, the language needs heavy editing; the phrasing sometimes makes it difficult to understand what the authors mean.

The language has been checked by an interpreter, which is confirmed by an additional statement. 

Line 47: change to " kept in environments as identical as possible"

The suggestion has been included.

Lines 59-60: "The animal's living environment...." this seems to be an unnecessary sentence

The sentence has been deleted.

Lines 66-67: "The research hypothesis assumed that the potential differences in the behaviour of chicks representing different breeds cannot result from the modifying impact of the environment." I suppose that is a possible hypothesis, but it seems very unlikely that behaviours from different breeds cannot be the result of modifying impact of environment- as the authors also say in the paragraph before.

This sentence has been reedited, as it had lost its most important meaning when translated into English. The intention was to express that the environment may not yet modify the behaviour of young chicks on the first days after hatching. Later, the birds' living environment will definitely modify their behaviour and it will be impossible to discriminate the effect of the environment from that of the breed. The possible breed-related differences (not those related to the environment) can only be checked in few-day-old chicks. The breeds, which we also observed in another experiment on adult birds, differed in behaviour but we did not know whether it was an effect of the breed or the genotype x race interactions. Therefore, we conducted the present experiments on chicks of these breeds.

The research hypothesis assumes that any differences in the behaviour between young chicks of different breeds cannot result from the modifying impact of the environment at such an early age.

Line 72-75: I don't understand the final sentence of the introduction.

We hope that this reedited paragraph is now understandable.

Line 97-98: "In total, 20-h recordings of 10-min tests and 36-h recordings of 30-min tests were analysed. " I don't understand this sentence.

As explained in the methodology, several tests lasted 10 minutes and the total recording time of these tests was 20 hours, while the second group of tests lasted 30 minutes and the total recording time was 36 hours.

Results: I am missing the absolute values of what was measured. It is unclear to me how the comparison values were calculated, and I would like to be able to see whether this is based on a few chicks performing the behavior or many. This makes it very difficult for the reader to interpret results. I am also skeptical about the statistical analysis, which from the tables appears to be one-on-one comparisons of the groups, without reporting overall effects of breed or age.

The tables have been reedited to make interpretation of the results easier. However, using absolute values of the traits in the tables will be misleading since, as written, the measured traits did not have a normal distribution. Therefore, after rank transformation that is used in such situations, the traits have different values and it is difficult to compare the values on the scale in which they were measured with values that were transformed.

Most nonparametric methods are based on taking the ranks of a variable and analyzing these ranks (or transformations thereof) instead of the original values. The NPAR1WAY procedure performs a nonparametric one-way analysis of variance. Other nonparametric tests can be performed by taking ranks of the data (using the RANK procedure) and using a regular parametric procedure (such as GLM or ANOVA) to perform the analysis.

In turn, using means after rank transformation instead of difference estimators can actually simplify interpretation of results. It is then easier to imagine the course of the experiment and its results; hence, the tables have been changed. The reviewer writes: "and I would like to be able to see whether this is based on a few chicks performing the behaviour or many" - there is no behaviour that would be based on a few chicks only. However, we want to point out that the study compared the breeds, and even if some behaviour was performed by a few chicks, this does not change the fact that there are or there are no differences between the breeds. There were 60 chicks in each breed and the tests were repeated, so it is not possible for any behaviour to be noted only several times. However, as we inform in the methodology, the traits did not have a normal distribution, which means that there was high variation. Providing the arithmetic mean at over 100% variabilities will be of little use.

As shown in the methodology section, two-factor models were used in the analyses, taking into account the effect of the breed and age:

"Multiple comparisons with the Bonferroni correction of the estimates of differences in the examined traits between the breeds were analysed in 2-factor models, taking into account the effect of the genetic group and the age of chicks at the time of the experiment”.

Discussion: as the results are difficult to interpret, the discussion is as well.

In this case, we cannot change anything, as we do not have specific guidelines. We hope that the reedited tables will make it easier to interpret the results

In general it appears that the authors have performed a considerable amount of work; I hope that in revision the paper will be more clear so that these results can be better interpreted.

Submission Date

22 December 2019

Date of this review

21 Jan 2020 16:27:17

Reviewer 2 Report

The subject of the present manuscript is very interesting and important especially considering animal welfare for poultry breeds in different husbandry systems. Used literature and materials and methods section can be improved though.

Abstract and summary:

The provided abstract and summary are identical except for some individual words. An abstract includes all the main aspects of the paper. A summary is a brief note that gives an overview, ideas and insight about major topics in a paper. So there are some differences expected between an abstract and a short summary. Results are too scarce in the abstract in my opinion.

Please carefully reconsider writing some words in capitals and when repeating the same words not.

Lines 31/32 Please explain competitiveness test I and II or write as in summary as two tests on competitiveness.

Line 47: Please explain the abbreviations for the used breeds here or in advance, since not every reader may conclude from the abstract.

Lines 47/48: What kind of different enrichment was preferred according to the authors? Please provide information.

Material and methods:

Line 76: Please change into animals, materials and methods or provide an extra rubric for the animals.

Line 83: Please provide more information about the used breeds and the differences occurring/expected in stress levels and temperament. And also about performance target, weight, plumage color.

Line 87: Please provide information about floor space per chick, for example: 100x70 cm sums up to 7000 cm2 and provides a 700 cm2 space/chick, if there weren’t any other facilities such as drinking and feeding troughs on the ground. The feeding troughs and drinker space needs to be subtracted.

Line 97: What kind of digital camera was used? (type, factory)

Table 1: For a better overview please change column 6 from its position to column position 2. The course of the experiment should stand at the beginning of the table. By observations (column 7): is this the definition of behavioural observations? If so please change from observation to definition.

Results:

Was there a preference of the chicks in daytime concerning the measurements?

Discussion:

Lines 198 and 214: won and winning doesn’t seem to be the proper scientific expression

Lines 205 and 206: see line 108, age was not taken into account?

Lines 215/217: based on what literature?

Line 257: based on what literature?

Lines 257-264: Literature?

In general:

Are the observations, measurements (points in time) and performed tests based on any literature? Please provide information. Why was described observation duration chosen? Why was the used age chosen? Laying hens are usually relocated at the age of 17/18 weeks of age. Why weren’t any tests performed between age 3 weeks (21 days) and the point of relocation? Or until plumage is fully developed (age 5/6 weeks)? Please provide a more detailed information about the housing conditions: illumination (lighting management, lux intensity, used lamps, flicker-free light?) information about bedding or wire-mesh floor? used feed (composition,  consistence, and ingrediences, ad libidum feeding or at least type and factory). Why was the effect of age not taken into account? (Table 2ff)

Author Response

Thank you for the review of the manuscript. Below we address each comment. Changes introduced in response to the review are marked in blue in the manuscript.

Open Review

English language and style

( ) Extensive editing of English language and style required 
( ) Moderate English changes required 
(x) English language and style are fine/minor spell check required 
( ) I don't feel qualified to judge about the English language and style 

Yes

Can be improved

Must be improved

Not applicable

Does the introduction provide sufficient background and include all relevant references?

( )

(x)

( )

( )

Is the research design appropriate?

(x)

( )

( )

( )

Are the methods adequately described?

( )

( )

(x)

( )

Are the results clearly presented?

( )

(x)

( )

( )

Are the conclusions supported by the results?

(x)

( )

( )

( )

Comments and Suggestions for Authors

The subject of the present manuscript is very interesting and important especially considering animal welfare for poultry breeds in different husbandry systems. Used literature and materials and methods section can be improved though.

Abstract and summary:

The provided abstract and summary are identical except for some individual words. An abstract includes all the main aspects of the paper. A summary is a brief note that gives an overview, ideas and insight about major topics in a paper. So there are some differences expected between an abstract and a short summary. Results are too scarce in the abstract in my opinion.

Please carefully reconsider writing some words in capitals and when repeating the same words not.

The suggestion has been included.

Lines 31/32 Please explain competitiveness test I and II or write as in summary as two tests on competitiveness.

The suggestion has been included.

Line 47: Please explain the abbreviations for the used breeds here or in advance, since not every reader may conclude from the abstract.

The suggestion has been included.

Lines 47/48: What kind of different enrichment was preferred according to the authors? Please provide information.

The information has been provided.

Material and methods:

Line 76: Please change into animals, materials and methods or provide an extra rubric for the animals.

We have changed it as suggested by the reviewer.

Line 83: Please provide more information about the used breeds and the differences occurring/expected in stress levels and temperament. And also about performance target, weight, plumage color.

The information about the breeds has been added. In turn, the information concerning stress and differences in the temperament was presented in detail in our previous studies [3,5] can be found there. However, we have supplemented the most important information in this field in the Introduction, where we write about the different breed-related preferences of the hens.

Line 87: Please provide information about floor space per chick, for example: 100x70 cm sums up to 7000 cm2 and provides a 700 cmspace/chick, if there weren’t any other facilities such as drinking and feeding troughs on the ground. The feeding troughs and drinker space needs to be subtracted.

After subtraction of the area of the pen's equipment, the floor space per chick was 0.021 m2, while the standard specified in the regulation is 0.017 m2/chick. This information has been added.

Line 97: What kind of digital camera was used? (type, factory)

We used a Panasonic HC-VXF990 camera. We did not provide this information in the manuscript to avoid advertising the product.

Table 1: For a better overview please change column 6 from its position to column position 2. The course of the experiment should stand at the beginning of the table. By observations (column 7): is this the definition of behavioural observations? If so please change from observation to definition.

We have changed it as suggested by the reviewer.

Results:

Was there a preference of the chicks in daytime concerning the measurements?

Each test was carried out for all chicks on one day, i.e. when the competitiveness test was performed on day 3, all chicks from the six pens were subjected to this test. Then, the same test was repeated on days 6, 9 ... etc. and all chicks were observed each time. This procedure was followed in all tests. Never were chicks from one pen tested on a different day than on that described in the procedures shown in Table 1.

Discussion:

Lines 198 and 214: won and winning doesn’t seem to be the proper scientific expression

The words have been replaced.

Lines 205 and 206: see line 108, age was not taken into account?

The sentence in line 108 was wrongly formulated, but the point was that the tables presented the results with reference to the age as well as the total results. The age was obviously included as a factor in the analyses, as written in the methodology: ”Multiple comparisons with the Bonferroni correction of the estimates of differences in the examined traits between the breeds were analysed in 2-factor models, taking into account the effect of the genetic group and the age of chicks at the time of the experiment”.

The sentence has been corrected to avoid misinterpretation.

Lines 215/217: based on what literature?

The information that motivation is the driving force of action was based on references [22,23]. However, the sentences questioned by the reviewer refer to our study results, but in fact they may have been vague. We hope that this issue will be clear now.

Line 257: based on what literature?

The information about the fact that the Zk breed is kept on organic farms and copes very well in such conditions is known to us from practice. There are papers on this subject from the 1960s, but written in Polish. Providing references that are generally unavailable to the modern reader is not effective. At the same time, the information about the breed is available e.g. on websites.

Lines 257-264: Literature?

The further sentences in this paragraph relate to the results of our study, therefore the Reviewer's comment is not clear to us.

In general:

Are the observations, measurements (points in time) and performed tests based on any literature? Please provide information. Why was described observation duration chosen? Why was the used age chosen? Laying hens are usually relocated at the age of 17/18 weeks of age. Why weren’t any tests performed between age 3 weeks (21 days) and the point of relocation? Or until plumage is fully developed (age 5/6 weeks)? Please provide a more detailed information about the housing conditions: illumination (lighting management, lux intensity, used lamps, flicker-free light?) information about bedding or wire-mesh floor? used feed (composition,  consistence, and ingrediences, ad libidum feeding or at least type and factory). Why was the effect of age not taken into account? (Table 2ff) 

The tests were based on studies conducted by other authors, but they were not identical - they must always be modified to the conditions in which they are carried out (e.g. Janczak, A. M., B. O. Braastad, and M. Bakken. "Behavioural effects of embryonic exposure to corticosterone in chickens." Applied Animal Behaviour Science 96.1-2 (2006): 69-82); 8.Janczak, A. M.; Heikkilä, M.; Valros, A.; Torjesen, P.; Andersen, I. L.; Bakken, M. Effects of embryonic corticosterone exposure and post-hatch handling on tonic immobility and willingness to compete in chicks. Appl Anim Behav Sci 2007, 107(3-4), 275-286).; . When designing the tests, we were based on research reported by other authors and our findings in pilot studies. When determining the time points, we intended not to make the chicks tired; we also wanted to make the elements added to the environment attractive to the chicks and to carry out the observation only as long as necessary to record the bird's response. At the same time, the analysis of longer recordings is time consuming. We had used 10-minute tests in a study of adult birds, and this time worked very well, e.g. Kozak et al., 2019a, b. The 30-minute tests consisted only of observing the natural behaviour of the chicks, which could last a long time. Besides, such observations are always longer than a specific test, e.g. competitiveness, where the chicks had eaten the treats within 10 minutes. The age was chosen in accordance with the assumption and hypothesis of the study. The chicks had to be very young to check whether the behavioural differences occurring in adult birds would be visible already after hatching. This rationale was presented in the Introduction and research hypotheses. Conducting experiments on older chicks would be pointless and inconsistent with the assumption of the study in this case. The older the chicks, the greater the environmental effect on their behaviour. Information about the conditions, e.g. lighting, temperature, etc. can be found in the methodology section:

One-day-old chicks were placed in 6 pens. There were 10 individuals of each breed in each pen. The group of birds within the pen did not change from the 1st to 23rd day of life. The only exception was one of the pens where a Leghorn chick was replaced on the 2nd day of life due to health condition. The pens with a surface area of 100cm x 70 cm and 60 cm high walls were kept in an experimental room at controlled temperature (22 oC) and humidity (68%). Each pen was heated with a 250W infrared heater ensuring a controlled temperature of the pen floor, i.e. 31 oC in the case of the one-day-old chicks. The temperature was gradually reduced to 23 oC on the 21st day of life. Each pen was equipped with two round drinkers with a 20-cm diameter and two 30-cm long feeders adapted for chicks. Complete diet and water were available ad libitum. The chicks were randomly selected for the experiment from a group of 500 chicks that had hatched at the Experimental Farm of the Scientific Institute. The birds were randomly allocated in the pens”

Standard feed for chickens of a certain age was used, but we cannot provide the name of its manufacturer or the exact composition, as these data are reserved for farms and commercial fee producers. The chicks are fed as required, which we specify in the paper. During the experiment, the chicks stayed in pens lined with substrate. It was replaced with a fresh and clean bedding once a day an hour before the start of the experiments.
As mentioned earlier, age was a factor taken into account in the statistical analyses.

Submission Date

22 December 2019

Date of this review

21 Jan 2020 11:40:32

Reviewer 3 Report

This study has found behavioural differences in chicks of alternative hen breeds. The methods seem properly applied, however the choice of pens of 10 animals should be justified. The statistical analysis should be better clarified (e.g. line 108, why? Repeated measures? Random effect of pen?). The practical implications of these findings should be better clarified.

Line 94: did chicks differ also for weight? were they weighted at the beginning of the experiment?

Table 1: it is not clear the starting time of each test during 9:00-16.00, the sequence of them, the time between one test and another.

Tables 2-7 are hardly readable, should be simplified to facilitate readers, and possibly reduced in number. Conversely, the number of observations should be included.

Discussion should be reduced, by removing speculations and parts that tend to lose the focus of the experiment (e.g. lines 181-183)

Author Response

Thank you for the review of the manuscript. Below we address each comment. Changes introduced in response to the review are marked in yellow in the manuscript.

Open Review

English language and style

( ) Extensive editing of English language and style required 
( ) Moderate English changes required 
(x) English language and style are fine/minor spell check required 
( ) I don't feel qualified to judge about the English language and style 

Yes

Can be improved

Must be improved

Not applicable

Does the introduction provide sufficient background and include all relevant references?

(x)

( )

( )

( )

Is the research design appropriate?

( )

(x)

( )

( )

Are the methods adequately described?

( )

(x)

( )

( )

Are the results clearly presented?

( )

(x)

( )

( )

Are the conclusions supported by the results?

(x)

( )

( )

( )

Comments and Suggestions for Authors

This study has found behavioural differences in chicks of alternative hen breeds. The methods seem properly applied, however the choice of pens of 10 animals should be justified. The statistical analysis should be better clarified (e.g. line 108, why? Repeated measures? Random effect of pen?). The practical implications of these findings should be better clarified.

The placement of 10 chicks of each breed (30 in total) in the pens was related solely to logistic reasons - a greater number of chicks would have been impossible to be thoroughly analysed in the recording, as this would require having a larger pen. The camera would not have covered the entire area and we would have had difficulty in heating the pen and ensuring proper conditions. In turn, a smaller number of chicks per pen would have been easier to analyse, but the number of pens should be increased. This was unfortunately impossible or could have resulted in failure to ensure appropriate conditions (e.g. no heating lamps). Line 108 did not only present the data presented in the results, including age and total values. It has been corrected to avoid misleading. It is difficult for us to improve the methodological part because there are no specific comments here. In our opinion, all relevant information related to statistical analyses was provided in the manuscript with information on the procedure and models used.

Line 94: did chicks differ also for weight? were they weighted at the beginning of the experiment?

Depending on the breed, the chicks weighed on average 35-37g. The chicks were weighed after hatching during assessment of their health. The body weight did not differ significantly between the breeds; therefore, this factor was not considered in further analysis. We have provided this information in the text.

Table 1: it is not clear the starting time of each test during 9:00-16.00, the sequence of them, the time between one test and another.

The tests did not start at 9 o'clock but were conducted between 9 a.m. and 4 p.m. This is the time when chicks are most active and the experimenters are in the workplace. The experiment was not conducted at the same hour deliberately to prevent conditioning of behaviour. We have supplemented this information in the text. By using different orders and different hours of testing, we intended to prevent the chicks from predicting what was to happen and from being able to "artificially" modify their behaviour. This unfortunately makes it difficult to answer the question about the time between one test and another, because this time was variable. It was never shorter than the total time of a single test for each group, i.e. the 10-minute test for all groups lasted a total of 60 minutes. Together with the time of changing and setting the cameras, it was at least 90 minutes (correspondingly longer for the 30-minute tests), which means that another experiment on this day could start the earliest after 90 minutes, but the intervals for the individual pens would be different due to the random mode of selection of the pens. However, when conducting the experiment, we had in mind the differences in time between the tests and, in this case, we used a principle that the next test could only be carried out when the chicks from all pens returned to their normal daily activity, i.e. when they were sleeping, approaching food or water, or exploring. The test was not applied when the chicks showed signs of tiredness e.g. the whole group was sleeping. That is why the experiment took such a long range of hours. This information has been added in the text.

Tables 2-7 are hardly readable, should be simplified to facilitate readers, and possibly reduced in number. Conversely, the number of observations should be included.

The tables have been reedited - the differences in the values ​​between the breeds have been removed and the means have been re-ranked (arithmetic means should not be provided due to the lack of a normal distribution of the traits, in which case the re-ranking procedure is applied). We hope that this approach will facilitate reading the data. Simultaneously, the significance level has been left in the table, as it shows whether the means between the breeds differ. We cannot reduce the number of tables because there were so many tests and results, otherwise we would have to give up some test results. Including the number of observations is rather not feasible. There were 60 chicks in each breed (this information is presented in the text); the observation, e.g. the number of chicks in the cage during the competitiveness test, was measured every 1 minute and the test was performed 7 times in each pen. Therefore, providing the number of observations would be equal to publishing the database. In studies where not only the number of individuals involved in the experiment but also the number of observations is important, a mean value is shown. Therefore, according to the reviewer's comment, we have provided re-ranked data, as written above. Absolute means have very high variability and are hence not suitable for interpretation.

Discussion should be reduced, by removing speculations and parts that tend to lose the focus of the experiment (e.g. lines 181-183)

This fragment has been removed, but the reviewer does not specify which information in the Discussion is debatable.

Submission Date

22 December 2019

Date of this review

16 Jan 2020 19:59:01

Round 2

Reviewer 3 Report

The authors have significantly improved the manuscript. However, I still struggle in reading the tables: they should be self-explainable and it remains unclear which is the meaning of the predictors and how they are calculated. Moreover, the original variables (minutes, number of events, birds etc. –mean and standard error or deviation) should be included.

Eg: Table 2. Estimators of differences in birds' behaviour traits in competitiveness/behaviour strategy tests in relation to the breed; significance of the differences and confidence interval.

What do the estimates represent? How are they obtained? Which is their unit of measure? Where is the confidence interval?

As regards the statistical section the authors wrote "The number of the pen where the chicks were kept was not a significant factor and was therefore not included in the analyses." I can imagine it was not, but my question was: why didn't you include the pen as a random effect in the analysis? This is the most suitable approach when comparing groups housed in different confinements.

Author Response

Thank you very much for the review. The new changes have been highlighted in green.

The authors have significantly improved the manuscript. However, I still struggle in reading the tables: they should be self-explainable and it remains unclear which is the meaning of the predictors and how they are calculated. Moreover, the original variables (minutes, number of

As we explained after the first reviews, the traits did not have a normal distribution. The variability of the traits is over 100%, and in this case, the absolute value of the traits is not suitable for interpretation. Comparison and interpretation of such means is highly difficult, as the variability of the trait should immediately be taken into account. To facilitate correct interpretation, we have used rank transformations and these results are presented in the tables, as described in the methodology section. Nevertheless, in order to provide the Reviewer with more detail, I give an example: the number of birds inside the cage (table 2 - Number of birds inside the cage (no.)) on day 3 was on average (without rank transformation) Lg = 1.4 chick, Sd = 1.3; Zk = 1.2, Sd = 1.1; Pb = 2.5 Sd = 1.6. Interpretation of the results with such a large deviation is difficult. In the table, the rank-transformed values ​​retain the same trend - the biggest number of Pb chicks and the lowest number of Zk birds. By the use of rank transformations the problem with the high variability and, therefore, difficult interpretation is eliminated. In the study, the value of the trait is not the key issue. It is not the number of heartbeats or another physiological index, which must be presented in the original scale, as its value itself is information. The present study is focused on comparison of the breeds, and it is not important whether there were 5 or 50 individuals in the cage. Instead, the information whether there were more individuals of the Lg, Zk or Pb breed is vital. The values ​​presented in the tables would not have any significance if they did not relate to the comparison of the breeds the information that there were 1.4 (or 574.17) Lg chicks in the cage is not meaningful and thus no conclusions can be drawn from this information. An important issue is the finding that they were significantly lower numbers of Lg than Pb or the same number as Zk. Therefore, the tables have not been supplemented with absolute means since, due to the absence of normal distribution, they would be more difficult to analyse than the transformed means.

Eg: Table 2. Estimators of differences in birds' behaviour traits in competitiveness/behaviour strategy tests in relation to the breed; significance of the differences and confidence interval.

What do the estimates represent? How are they obtained? Which is their unit of measure? Where is the confidence interval?

Certainly, the table caption is incorrect, as well as in the other tables. The content of the tables has been changed in accordance with the comments contained in the first review, and the signatures of the tables have not been changed due to our oversight, for which we apologise. In fact, the tables present the mean, rank-transformed value of measured traits (Estimate), and the significance of differences between the means for the breeds (Pr > |t|). The units of measurement were included in the description of Table 1 but we have supplemented this information in each table to erase any doubt.

As regards the statistical section the authors wrote "The number of the pen where the chicks were kept was not a significant factor and was therefore not included in the analyses." I can imagine it was not, but my question was: why didn't you include the pen as a random effect in the analysis? This is the most suitable approach when comparing groups housed in different confinements.

Before the final analyses, all possible factors were verified in terms of significance. We also checked the impact of the pen as a random factor. However, it turned out that it was not a significant factor, therefore we did not include it in the analyses. Perhaps the room is rarely an insignificant factor, but in this experiment we used were boxes in fact located in one room, all next to each other, in identical conditions.